# Understanding the Role of Soft X-ray in Charging Solid-Film and Cellular Electrets

**DOI:** 10.3390/nano12234143

**Published:** 2022-11-23

**Authors:** Yue Feng, Zehong Rao, Ki-Young Song, Xusong Tang, Zilong Zhou, Ying Xiong

**Affiliations:** 1School of Mechatronical Engineering, Beijing Institute of Technology, Beijing 100081, China; 2Shanghai Electro-Mechanical Engineering Institute, Shanghai 201109, China; 3Laboratoire Catalyse et Spectrochimie, ENSICAEN, Université de Caen, CNRS, 6 Boulevard Maréchal Juin, 14050 Caen, France

**Keywords:** electret, cellular electret, soft X-ray, charge density

## Abstract

Solid-film electrets and cellular electrets are defined as promising insulating dielectric materials containing permanent electrostatic and polarizations. High-performance charging methods are critical for electret transducers. Unlike dielectric barrier discharge (DBD) charging, the soft X-ray charging method, with its strong penetration ability, has been widely used in electrets after packaging and has even been embedded in high-aspect-ratio structures (HARSs). However, the related charging model and the charging effect of the soft X-ray irradiation remain unclear. In this study, the charge carrier migration theory and the one-dimensional electrostatic model were employed to build the soft X-ray charging models. The influence of soft X-ray irradiation under deferent poling voltages was investigated theoretically and experimentally. The conducted space charge measurement based on a pulsed electro-acoustic (PEA) system with a soft X-ray generator revealed that soft X-ray charging can offer higher surface charge densities and piezoelectricity to cellular electrets under the critical poling voltage lower than twice the breakdown voltage.

## 1. Introduction

Permanently charged electrets with a solid film [1,2] or cellular structure [3,4] are insulating dielectric materials that exhibit a net quasi-permanent electrical charge or dipole moment. These electrets carrying trapped charges with strong electrostatic or piezoelectric effects have been widely used in microsystems such as pressure or tactile sensors [5], accelerometers [6], energy harvesters [7,8,9], and actuators [10].

The first electrets were made from natural organics such as carnauba, paraffin, and rosin. However, the low charge densities and poor thermal stabilities of such organic electrets limited their engineering applications. The developed electrets can be classified into inorganic electrets and polymer electrets. Inorganic electrets such as SiO_2_ and Si_3_N_4_ have relatively high charge densities and good compatibility with microelectromechanical system (MEMS) processes. However, the trapped charge of inorganic electrets can easily leak out, resulting in poor charge stability [1,2]. In contrast, polymer electrets, especially fluoropolymer electrets such as polytetrafluoroethylene (PTFE), fluorinated ethylene propylene (FEP), and CYTOP^TM^, have been widely used in electrostatic transducers because of their high mechanical flexibility, excellent charge stability, and high dielectric breakdown strength [2]. In addition, it was found that the polymer electrets doped with nanoparticles can effectively enhance charge density [11]. Some natural biological tissues such as bone, blood, and skin also exhibit electret effects in the process of human life. These tissue electrets, together with synthetic biocompatible polymers such as PTFE and FEP, are known as bioelectrets [2] that can regulate neural signals, thinking processes, regeneration of biological memory, etc. Recently, the biodegradable electrets [12] represented by polylactic acid (PLA) have attracted interest due to their potential to replace some non-fluorine polymer electrets such as polyethylene (PE) and polypropylene (PP).

Electret films can be prepared by hot pressing or spin coating. In order to precisely control the morphology of the electret at the microscale, MEMS processes, such as chemical vapor deposition (CVD) and etching, are often used to form patterned electrets [10,13,14]. In addition, 3D printing, as a low-cost and rapid manufacturing method, can also print electrets of low melting temperature on flexible substrates [15].

The performance of electret transducers can be determined from the density of charge build-up on the surface layers or internal air voids. High-performance charging methods in the regulation of trapped charges have received great interest. Corona charging [16,17], as a classic unipolar in-air charging method, can easily achieve a high surface charge density on electret surfaces. However, it is incapable of charging electrets after packaging and when embedded in high-aspect-ratio structures (HARSs) because corona ions cannot pass through obstructive substrates and narrow gap openings [13,14,18,19]. The electron beam charging method [20,21] enables a large charging area coverage, as well as localized patterning, but it requires a vacuum environment, and only monopolar charges are deposited. Dielectric barrier discharge (DBD) technology [22,23], which relies on the Paschen breakdown in the air voids caused by an external direct-current (DC) voltage, enables hetero charges to deposit on the internal voids of cellular electrets. However, the inevitable extinguishment of breakdown stops charge transportation, eventually reducing the charging efficiency [22]. To address those issues, Suzuki et al. [13,14,18,19] developed a prospective soft X-ray charging technique in air and revealed the fundamental charging mechanism. Because of the photoionization effect and penetration ability, soft X-rays can be easily transmitted through substrates and generate large amounts of ions in narrow voids, which is rather effective for charging packaged electrets and cellular electrets, even when embedded in HARSs. However, the specific charging models of solid-film and cellular electrets with soft X-ray irradiation remain unclear. In addition, insufficient in situ charge measurements have been provided to demonstrate the charging behaviors of cellular electrets. These aspects are essential to theoretically and experimentally clarify and take full advantage of soft X-ray charging technology.

In this work, we propose a mathematical model of soft X-ray charging to explain the deposition of X-ray irradiated ions on the surface of solid-film electrets and the void surfaces of cellular electrets. The electrification mechanism of soft X-ray charging was interpreted according to the charge carrier migration theory in air dielectric and the one-dimensional electrostatic model of the charged electrets. Importantly, pulsed electro-acoustic (PEA)-based techniques, such as in situ space charge measurement, were employed to describe the charging behaviors of soft X-ray charged cellular electrets. The figures of merit, such as the piezoelectric constant *d*_33_ and surface charge density *σ*, are provided to demonstrate the charging superiority in the comparison between soft X-ray and DBD charging methods.

## 2. Theoretical Modeling of Soft X-ray Charging Method

### 2.1. Charging Solid-Film Electrets

Soft X-rays represent short-wavelength (1–10 Å) electromagnetic radiation with photon energies typically below 10 keV, whose interactions with matter are dominated by the photoelectric effect [24]. Figure 1a depicts the mechanism of soft X-ray polarizing electrets, where high-initial-kinetic-energy X-ray photons collide with gas atoms to generate electron–ion pairs as the soft X-rays are irradiated in the air. By applying a high DC as a biased electric field between the top and bottom electrode plates, the positive and negative ions are attracted to the oppositely charged electrodes, and unipolar charges are subsequently transferred to the electret surface.

Soft X-rays are conventionally generated through the processes of field-emission (FE) electron impact on a metal target. The initial X-ray intensity (*I*_0_) in the transmitting window of the soft X-ray tube can be calculated using an empirical equation (Equation (1)) [25]:(1)$$I_{0}=\eta i_{X}V_{X}{}^{2}Z[J/s\cdot m^{2}]$$
where *η* ≈ 1.1–1.4 × 10^−9^/V is the proportionality coefficient, *Z* is the atomic number of the target metal, and *V_X_* and *i_X_* are the tube voltage and tube current of the X-ray generator, respectively.

As illustrated in Figure 1b, the soft X-rays pass through three layers of air (distance *L*), a top electrode (thickness *d*), and another air layer (distance *H*). According to the X-ray attenuation law [26], the X-ray intensity variation (Δ*I*) due to absorption by air in the gap can be written as:(2)$$\Delta I=I_{up}-I_{down}=I_{0}\exp\left(-\mu_{air}L-\mu_{e}d\right)\left[1-\exp\left(-\mu_{air}H\right)\right]$$
where *μ_air_* and *μ_e_* are the linear absorption coefficients of X-rays in air and the top metal layer, respectively. The amount of electron–ion pairs (*n_t_*) generated between the top and bottom electrodes per unit time can be written as:(3)$$n_{t}=\frac{\Delta I\cdot S}{W}=\frac{I_{0}\cdot S\cdot \exp\left(-\mu_{air}L-\mu_{e}d\right)\left[1-\exp\left(-\mu_{air}H\right)\right]}{W}$$
where *S* is the electret area, and *W* = 33.7 eV is the average ionization energy of the air gas.

The charging current *J* formed by the directional movement of electrons under the action of the external electric field can be described by the migration–diffusion equation [27]:(4)$$J=en_{t}bE_{g}+eD_{i}\frac{\partial n_{t}}{\partial x}$$
where *e* is the fundamental charge, *D_i_* is the diffusion coefficient, *v_d_* = *bE_g_* is the driving speed of charge carriers (ions and electrons) under electric field *E_g_*, and the carrier mobility (*b*) determined by *E_g_* obeys the power law [28]. The *E_g_* in the air gap can be expressed as:(5)$$E_{g}=\frac{\varepsilon_{0}\varepsilon_{r}V_{\mathrm{DC}}-\sigma_{S}h}{\varepsilon_{0}\varepsilon_{r}H+\varepsilon_{0}h}$$
where *ε*_0_ is the vacuum permittivity, and *σ_S_*, *ε_r_*, and *h* are the surface charge density, relative permittivity, and thickness of the electret, respectively. The variation in surface potential (*V_S_*) with charging time (*t*) can be calculated with respect to the charging current (*J*):(6)$$\frac{dV_{S}(t)}{dt}=\frac{h}{\varepsilon_{0}\varepsilon_{r}}\frac{d\sigma_{S}(t)}{dt}=\frac{h}{\varepsilon_{0}\varepsilon_{r}}J(t)$$

The soft X-ray photoionized charges continuously accumulate on the electret surface, and the effective electric field (*E_g_*) approaches zero. This charge deposition process ends after a sufficient charging time. According to Equations (5) and (6), the amplitude of the maximum surface potential (*V_S_*_,max_) of the electret surface is equal to that of the applied bias voltage *V*_DC_.

### 2.2. Charging Cellular Electrets

Figure 2a gives a schematic diagram of a cellular electret with internally charged voids. The electromechanical operation of cellular electrets is described by a one-dimensional model first proposed by professor Sessler [3]. As described in Figure 2b, a cellular electret consists of *N* solid electret layers and (*N* − 1) air layers in thickness *h_p_*_,*i*_ and *h_a_*_,*i*_, respectively. The electric fields of the *i*th electret and void are *E_p_*_,*i*_ and *E_a_*_,*i*_, respectively. The permanent charges are assumed to locate only on the void–electret interfaces, and the charges at opposite sides of each void are equal in magnitude when a poling voltage (*V*_DC_) is applied.

Based on Gauss’s theorem and Kirchhoff’s second law, the *E_a_*_,*i*_ in the *i*th air void can be expressed as [22]:(7)$$E_{a,i}=\frac{\varepsilon_{0}\varepsilon_{r}V_{\mathrm{DC}}-\sigma_{i}\sum_{i}^{N}h_{p,i}}{\varepsilon_{0}\sum_{i}^{N}h_{p,i}+\varepsilon_{0}\varepsilon_{r}\sum_{i}^{N-1}h_{a,i}}$$

In the soft X-ray charging process, charge accumulation on the surfaces of the air voids is nearly terminated, and the strength of the electric field is insufficient to separate ions (i.e., *E_a_*_,*i*_ = 0). The saturated surface charge density (*σ_i_*) in the *i*th void can be written as:(8)$$\sigma_{i}=\frac{\varepsilon_{0}\varepsilon_{r}V_{\mathrm{DC}}}{\sum_{i}^{N}h_{p,i}}$$

It is clear that *σ_i_* cannot increase further when *V*_DC_ is decreased. In addition, the well-known back discharge effect [22] occurs while *E_a_*_,*i*_ formed by the trapped charges is higher than the Paschen breakdown field (*E*_DBD_). Thus, the maximum surface charge density (*σ_i_*_,max_) is derived as:(9)$$\sigma_{i,\max}=\frac{E_{\mathrm{DBD}}(\varepsilon_{0}\sum_{i}^{N}h_{p,i}+\varepsilon_{0}\varepsilon_{r}\sum_{i}^{N-1}h_{a,i})}{\sum_{i}^{N}h_{p,i}}$$

The poling voltage to achieve *σ_i_*_,max_ is numerically equal to the voltage (*V*_DBD_) at which DBD discharge occurs in the *i*th air void without soft X-ray irradiation:(10)$$V_{\mathrm{DBD}}=E_{\mathrm{DBD}}(\sum_{i}^{N}h_{p,i}/\varepsilon_{r}+\varepsilon_{0}\varepsilon_{r}\sum_{i}^{N-1}h_{a,i})$$

The *E*_DBD_ of the *i*th void can be written as follows:(11)$$E_{\mathrm{DBD}}=\frac{Ap}{B+\ln\left(ph_{a,i}\right)}$$
where *p* is the pressure of the air void, and parameter *B* = ln[*C*/ln(1 + 1/*γ*)]. The constants *A* and *C* depend on the composition of the gas, and *γ* is the second ionization coefficient.

The one-dimensional model of the cellular electret can be further simplified to a sandwich structure model with one air layer and two electret layers, where $h_{p}=\sum_{i=1}^{N}h_{p,i}/2$ and $h_{a}=\sum_{i=1}^{N-1}h_{a,i}$. The absorption of X-rays in the electret layer can be ignored, and the X-ray ionization rate (*n_t_*′) in the air layer can be written as Equation (12). *n_t_*′ and *E_a_* were used to calculate the charging current (*J*) and the surface charge density versus time on the internal void surfaces:(12)$$n_{t}^{\prime }=\frac{I_{0}\cdot S^{\prime }\exp\left(-\mu_{air}L^{\prime }-\mu_{e}d_{e}-\mu_{p}h_{p}\right)\left[1-\exp\left(-\mu_{air}h_{a}\right)\right]}{W}$$
where *μ_p_* is the linear absorption coefficients of X-rays in the polymer layer.

Regarding the poling voltage, the theoretical saturation surface charge density in soft X-ray charging can be derived as:(13)$$\sigma_{X\text{-}\mathrm{model}\text{-}\mathrm{saturation}}=\{\begin{array}{l} \varepsilon_{0}\varepsilon_{r}V_{\mathrm{DC}}/2h_{p}\\ \varepsilon_{0}\varepsilon_{r}V_{\mathrm{DBD}}/2h_{p} \end{array}\begin{array}{l} 0\leq V_{\mathrm{DC}}<V_{\mathrm{DBD}}\\ V_{\mathrm{DC}}\geq V_{\mathrm{DBD}} \end{array}$$

Equation (14) is used to describe the surface charge density dependence on poling voltage in the DC DBD charging [4,22,23]:(14)$$\sigma_{\mathrm{DBD}\text{-}\mathrm{model}}=\{\begin{array}{lll} 0 & & 0\leq V_{\mathrm{DC}}<V_{\mathrm{DBD}}\\ \varepsilon_{0}\varepsilon_{r}\left(V_{\mathrm{DC}}-V_{\mathrm{DBD}}\right)/2h_{p} & & V_{\mathrm{DBD}}\leq V_{\mathrm{DC}}<2V_{\mathrm{DBD}}\\ \varepsilon_{0}\varepsilon_{r}V_{\mathrm{DBD}}/2h_{p} & & V_{\mathrm{DC}}\geq 2V_{\mathrm{DBD}} \end{array}$$

According to Equations (13) and (14), there is no charge accumulation on the void interfaces at *V*_DC_ < *V*_DBD_ in DBD; in contrast, soft X-ray charging can offer a much higher surface charge density at *V*_DC_ < 2*V*_DBD_. Furthermore, a much lower poling voltage (*V*_DC_) is required to achieve *σ*_max_ in the soft X-ray charging (i.e., *V*_DC_
*= V*_DBD_).

## 3. Materials and Methods

A commercial soft X-ray generator with a tube voltage of 11.0 kV (SXN-10H, SUNJE electronics, Busan, Republic of Korea) was used to create a high-density plasma. A charging voltage supplier with a large amplitude (−30 to 30 kV) was used to provide a high-level biased electric field between a 15.0 μm-thick aluminum foil as a top electrode and a copper bottom electrode plate with a spacing of 2.0 cm.

A solid-film PTFE electret specimen with a thickness of 500 μm (area of 20 mm × 20 mm) was charged by soft X-ray in the air. Its surface potential was measured with a commercial electrostatic voltmeter (Model 279, Monroe Electronics, Lyndonville, NY, USA). To experimentally describe the charge density on void surfaces of cellular electrets, we employed an in situ space charge measurement method based on the pulsed electro-acoustic (PEA) technique [29,30,31,32] for the cellular electret measurement.

The PEA method is often used to detect charge distribution in solid dielectrics. Figure 3a shows a PEA device for in situ space charge measurement of soft X-ray-charged cellular electrets. Different from traditional PEA devices, our system had the capability of measurement with soft X-ray irradiation. The measurement mechanism was based on the detection of the acoustic waves generated by the vibration of charges inside the specimen after a pulse voltage (pulse width of 20 ns; pulse amplitude of 1400 V) was applied. The generated acoustic waves were detected by a 28 μm-thick polyvinylidene difluoride (PVDF) piezoelectric transducer (LDT0-028K, TE Connectivity, Middletown, CT, USA) and then amplified by a 64 dB-gain low noise amplifier, where the intensity of the signal represented the charge density, and the arrival time of the acoustic wave was used to confirm the positions of the charges inside the specimen.

It should be noted that there is a serious barrier effect [33] of air layers in the process of measuring cellular electrets using PEA systems, which is associated with strong reflections of acoustic waves caused by a large difference in acoustic impedance (*Z*) between air and electret layers (e.g., *Z_air_* = 432 kg/m^2^s and *Z_PTFE_* = 3.15 × 10^6^ kg/m^2^s). Hence, the prepared cellular electret specimen in the present work only contains a single void rather than multiple voids to ensure a clear signal of charge density distribution. The cellular PTFE specimen comprised a 52.3 μm-thick hollow polyethylene terephthalate (PET) film as a void supporter and two 100 μm-thick PTFE membranes. The voided PTFE electret with 15 μm-thick Al electrode layers had a total thickness of 282.3 μm and an area of 10 mm × 10 mm.

The propagation behaviors of the acoustic waves generated by deposited charges on interfaces are shown in Figure 3b to better understand how the PEA measurement works on cellular electrets. The acoustic waves’ transmission time (*τ*) in different layers can be calculated as *τ_i_ = h_i_*/*v_i_*, where *v_i_* is the sound velocity of related materials (e.g., *v_PTFE_* = 1350 m/s in PTFE layers, and *v_air_* = 344 m/s in air layers). Here, the moment at which the deposited charges in the PTFE–Al interface resulted in the acoustic wave (marked with *I*, blue lines in Figure 3b,c) reaching the PVDF sensor is set to 0 ns. After an elapsed time of 74.1 ns, the second wave (marked with *II*, red lines in Figure 3b,c) arrived due to charge accumulation in the PTFE–air interface. Before receiving the subsequent third wave (wave *III* associated with the air–PTFE interface, purple lines in Figure 3b,c), two additional waves of non-interest (reflected wave *I′* and wave *II′*) arrived successively. Unfortunately, wave *III* and reflection wave *II′* had a very short time interval (i.e., 3.8 ns), making it difficult to distinguish wave *III* from the other complex signals. The calculated results are consistent with the experimental results, as shown in Figure 3c. Here, we manually flipped the specimen to obtain clear signals for wave *III* and wave *IV*, avoiding the interference of the reflected signals.

## 4. Results and Discussion

### 4.1. Solid-Film PTFE Electret

The X-ray ionization rate (*n_t_*)-dominated soft X-ray efficiency is mainly influenced by the thickness (*d*) and linear absorption coefficient (*μ*) of the top electrode layer. In our facility, the tube voltage (*V_X_*), current (*i_X_*), and photon energy of the commercial soft X-ray generator using a beryllium (Be) metal target were 11 kV, 200 μA, and 8 keV, respectively. The distance (*H*) between the top and bottom electrode layers was set to 2 cm, and the X-ray tube was positioned 3 cm above the top electrode layer.

The curves of *n_t_*, which were dependent on the thickness of the top electrode layer and calculated from Equations (1)–(3), are depicted in Figure 4a, showing an exponential decay trend. The differences in atomic number and physical density of the aluminum and copper electrodes resulted in different linear absorption coefficients (*μ*) (i.e., *μ*_Al_
*=* 135.9 cm^−1^ in aluminum and *μ*_Cu_ = 470.8 cm^−1^ in copper), where a lower *μ* means higher X-ray transmission ability through electrode layers with the same thickness, which eventually leads to a higher *n_t_* beneath the top aluminum electrode layer.

The numerical relationship between the surface potential (*V_S_*) and the X-ray ionization rate (*n_t_*) can be obtained from Equations (4)–(6). However, it is difficult to confirm the specified parameters of the field-dependent carrier mobility (*b*). In this case, experimental data are used to obtain a fitted expression *b = kE^n^* of the specimen, where *k* and *n* are constants and exponents without physical meaning, respectively. Numerous studies indicated that the properties of the electret, such as dielectric constant, conductivity, thickness, and the ability to trap the ion species, have a great impact on the charging results [11,34,35,36].

Usually, negatively charged electrets tend to have higher surface potential than positively charged electrets under the same conditions [36,37] because some of the electrons cannot attach to air molecules while the charging polarity is negative, and the lightweight electrons are much easier to inject into the specimen rather than on the surface. To reflect the ability of the soft X-ray charging method, the negative-polarity results (blue dots in Figure 4b) were used to fit the mobility (*b*) as $b\left(m^{2}V^{-1}s^{-1}\right)=7.5E_{g}^{\left(0.3-0.15logV_{\mathrm{DC}}\right)}\times 10^{-3}$. It is clear that the *V_S_* of the positively charged specimen has several deviations from the fitted curve using the negative-polarity data.

Figure 5a,b shows the variation in the surface potentials (*V_S_*) with the charging time versus poling voltages (*V*_DC_). The prediction using the fitted model matches the experimental results well. The *V_S_* changed sharply at the beginning and gradually reached saturation with increased charging time. At *V*_DC_ = ±0.5 kV, ±1.0 kV, ±1.5 kV, and ±2.0 kV, the saturation surface potential *V_S_*_,max_ was about ±440 V, ±880 V, ±1350 V, and ±1700 V, respectively. The *V_S_*_,max_ could not reach *V*_DC_ probably because the electric field (*E_g_*) was not sufficient to drive charge injection into the electret as the *V_S_* increased. Thus, *V_S_* could not further increase, even though *E*_g_ had not yet dropped to zero. In addition, the saturation charging time at *V*_DC_ = ±0.5 kV, ±1.0 kV, ±1.5 kV, and ±2.0 kV was about 7.5 s, 15 s, 25 s, and 32.5 s, respectively.

### 4.2. Cellular PTFE Electret

It is important to experimentally verify the relationship between the number (*N*) of electret layers and the surface potential (*V_S_*_,*i*_) of each electret layer for the soft X-ray-charged cellular PTFE electret. The specimens with *N* = 2 and 3 were charged at *V*_DC_ = 1.0 kV for the same charging time of 10 min, as well as the same *h_p_*_,*i*_ = 100.0 μm and *h_a_*_,*i*_ = 1.2 mm. After charging, each electret layer was purposely separated from the assembled cellular electret and measured by Model 279. Figure 6 shows that each electret layer had almost the same surface potential (*V_S_*_,*i*_), which was approximately equal to *V*_DC_/*N*. For example, the initial *V_S_* of surfaces *S*1, *S*2, *S*3, and *S*4 were about 325 V, −335 V, −330 V, and 320 V, respectively. For the specimen with *N* = 3, the decay of the deposited hetero-charges showed a relatively good stability for both positive and negative charges within the initial 7 days.

PEA-measured space charge profiles of the soft X-ray-charged cellular PTFE electret with *h_p_* = 100.0 μm and *h_a_* = 52.3 μm after a charging time of 20 min are shown in Figure 7. Measurements of space charges were carried out after a 5 min short-circuit period. The appearance of hetero-charges density peaks located at the air–PTFE and PTFE–Al interfaces correspond to trapped charges due to the photoionization of soft X-ray-irradiated air. The charge densities continuously increased with increasing *V*_DC_ and reached saturation when *V*_DC_ was above 3.0 kV.

The measured surface charge density (*σ*_PEA_) at the air–PTFE interface was derived from the integral of the PEA-measured space charge density, *ρ*(*x*), at the interface region [38]. Figure 8a depicts *σ* below the poling voltages *V*_DC_. For a charging time of 20 min, the soft X-ray-charged specimen (circles on solid lines) exhibited an approximately linear increase in *σ*_PEA_, and eventually a maximum (*σ*_max_ = ~0.18 mC/m^2^) was observed when *V*_DC_ was over 3 kV. Compared with the DBD charging (*σ*_DBD-model_, blue dotted line), soft X-ray based cellular electret charging technology was able to generate high-density plasma and enabled higher surface charge densities at the air–PTFE interface, which typically exceeded several times the value in DBD-based charging when *V*_DC_ < 2 *V*_DBD_. Although both *σ*_PEA_ and *σ*_DBD-model_ can eventually achieve the same *σ*_max_, the critical poling voltage (*V*_DC_) required to reach *σ*_max_ in soft X-ray was 1.3 *V*_DBD_, which meant the soft X-ray charging required a much lower *V*_DC_ to achieve charging saturation.

Figure 8b illustrates the variation in the surface charge densities (*σ*) on the top and bottom electret surface layers with the charging time at *V*_DC_ = 1.0 kV; *σ* increased rapidly within the first 2 min, and the growth rate gradually slowed down with charging time. The negatively charged top electret surface had a slightly higher charge density than that of the bottom surface. The experimental and the calculated surface charge densities are in good agreement within a short charging time, but obvious differences were observed after 3 min, which can be attributed to insufficient photoionized gas inside the enclosed cellular structure. In addition, as discussed previously, it is impossible to reach the theoretical saturation (*σ*) because charge carriers require a sufficient electric field to be injected into the electret layer.

The piezoelectric constant *d*_33_—as a figure of merit to indicate the piezoelectric response of cellular electrets—was given with respect to the surface charge density. The theoretical value of *d*_33_ was determined in References 4, 14, 22, and 23. The effective Young’s modules (*Y*) of the cellular PTFE electret was estimated to be 1.3 MPa. The quasi-static method is a direct method to assess *d*_33_. As shown in Figure 9, a lightweight pre-load was used to hold the cellular electret specimen, and external force was applied or removed along the thickness direction of the specimen through a mass (*M*) of 20.05 g. The amount of charges (*Q*) generated by the specimen was determined using a fabricated charge meter (INA 128, Texas Instruments, USA). The relationship between quasi-static *d*_33_ and generated *Q* can be expressed as *Q = Mg*·*d*_33_. The measurement results of quasi-static *d*_33_ (circles, solid line shown in Figure 9) describe a similar tendency as the surface charge density versus poling voltages. The obtained maximum *d*_33_ was about 150 pC/N. This finding further verifies that the soft X-ray charging method can achieve a high piezoelectricity at a lower charging voltage (*V*_DC_) (e.g., *V*_DC_ < 2 *V*_DBD_).

## 5. Conclusions

In this paper, charging models and charging behaviors are analyzed with respect to soft X-ray charging for both solid-film and cellular electrets. The soft X-ray charging model, associated with the electrostatic model, is proposed based on the theories of X-ray photoionization and charge carrier migration diffusion. Well-fitted models are useful to predict the charging preference and understand the role of soft X-ray in charging. Importantly, the surface charge density on the internal voids of the cellular electret was measured in situ using a PEA-based system with soft X-ray irradiation. The soft X-ray-charged cellular electrets required a much lower poling voltage to achieve saturation than that in the DBD charging method. The critical poling voltages of the maximum surface charge density and piezoelectricity were 1.3 *V*_DBD_ and 2 *V*_DBD_ for soft X-ray charging and DBD charging, respectively. These findings are significant to better understanding the mechanism and application of the soft X-ray charging method in electrets and electret transducers.

## Figures and Tables

**Figure 1 nanomaterials-12-04143-f001:**
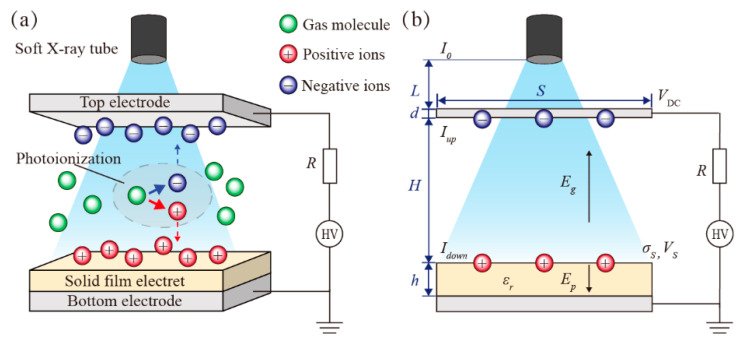
Soft X-ray charging solid-film electrets: (**a**) schematic of dynamic behavior of ions; (**b**) equivalent electrical model.

**Figure 2 nanomaterials-12-04143-f002:**
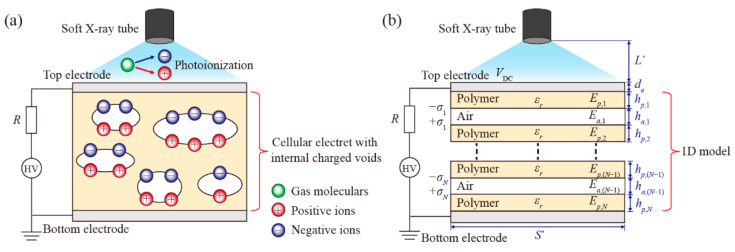
Soft X-ray charging cellular electrets. (**a**) Schematic of cellular electret; (**b**) One-dimensional model.

**Figure 3 nanomaterials-12-04143-f003:**
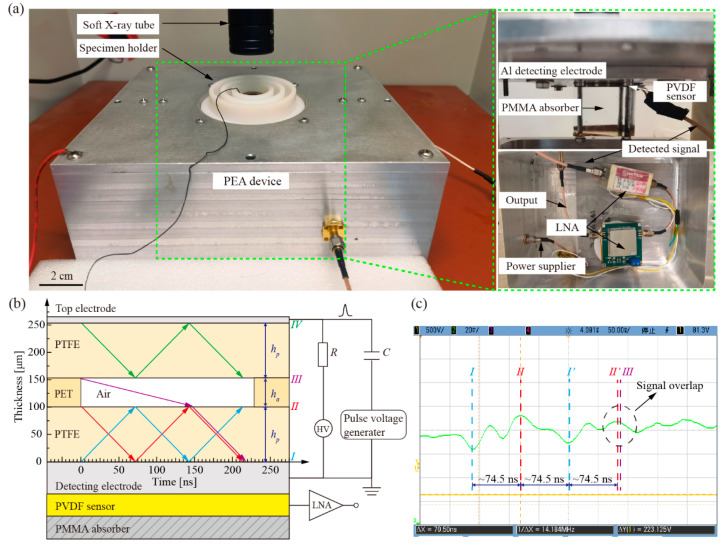
In situ PEA measurement of the space charge for X-ray irradiated cellular electret. (**a**) Experimental setup of PEA system. (**b**) Calculated acoustic waves transmission in the prepared cellular PTFE electret. (**c**) Electrical response signal received by a PVDF sensor.

**Figure 4 nanomaterials-12-04143-f004:**
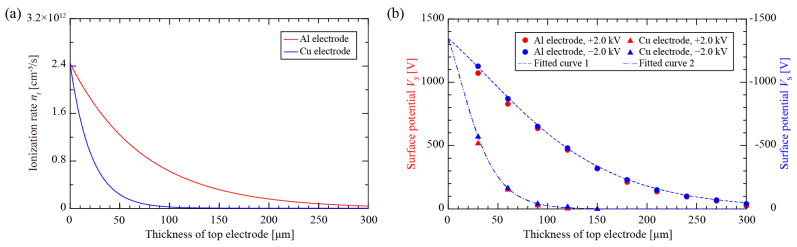
The metal materials influenced the soft X-ray ionization rate and the charging performance: (**a**) ionization rates beneath the electrode layer; (**b**) obtained surface potentials after a charging time of 10 s at *V*_DC_ = ±2.00 kV.

**Figure 5 nanomaterials-12-04143-f005:**
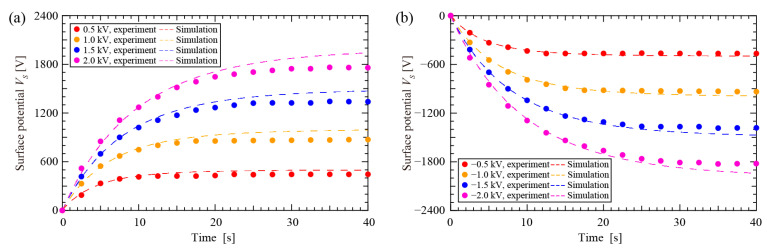
Surface potential of soft X-ray charging PTFE electret. (**a**) Positive charge deposition; (**b**) Negative charge deposition.

**Figure 6 nanomaterials-12-04143-f006:**
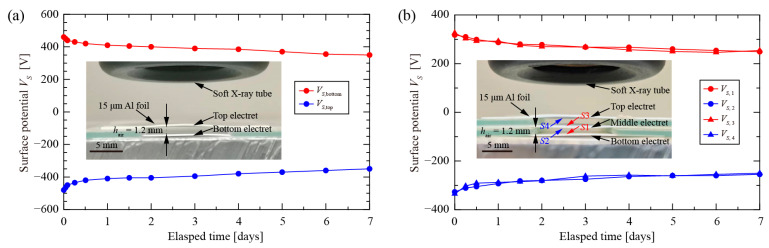
Surface potential of each electret layer inside cellular PTFE electret after soft X-ray charging at *V*_DC_ = 1.00 kV. (**a**) Specimen with *N* = 2; (**b**) Specimen with *N* = 3.

**Figure 7 nanomaterials-12-04143-f007:**
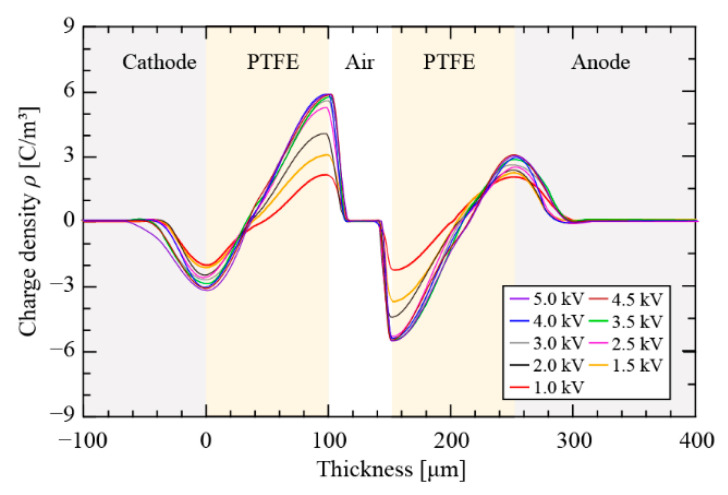
Measured space charge distribution of cellular PTFE electrets.

**Figure 8 nanomaterials-12-04143-f008:**
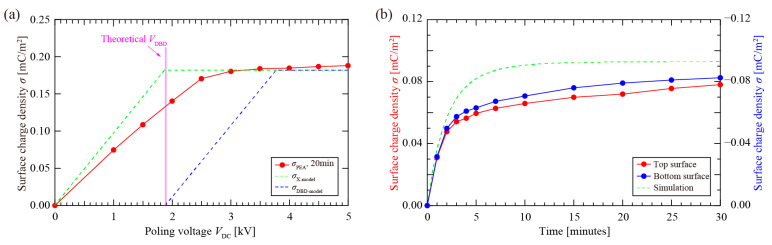
Surface charge densities (*σ*) of soft X-ray-charged cellular electret. (**a**) The dependence of *σ* on the poling voltage (*V*_DC_); (**b**) The dependence of *σ* on charging time at *V*_DC_ = 1.00 kV.

**Figure 9 nanomaterials-12-04143-f009:**
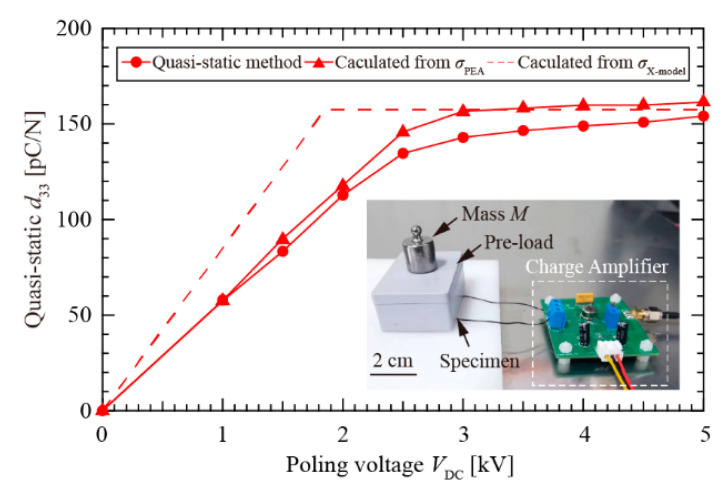
The dependence of quasi-static *d*_33_ on the poling voltage (*V*_DC_) in cellular PTFE electrets.

## Data Availability

Data are contained within the article.

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
