# Peer review of "Understanding the Role of Soft X-ray in Charging Solid-Film and Cellular Electrets"

_nanomaterials, 2022, doi:10.3390/nano12234143_

Round 1

Reviewer 1 Report

The work is relevant to the development of technology of electrets and the presented experimental methodology is useful for the purpose.
Comments:
1.    Some performance issues with electrets should be mentioned in the Introduction such as charge stability, depolarization.
2.    Advancements in electret fabrication using novel manufacturing technologies (3D printing) should be shortly described in the Introduction.
3.    Several typical and emerging electret polymeric materials should be shortly mentioned in the Introduction including biopolymers, biodegradable materials. It is not explained why PTFE is selected for the study because there is a need to study more sustainable materials for electret-based transducer fabrication.  
4.    There is no reference given to eq. 7. If it was derived by authors, then the derivation should be provided.
5.    Line 148. It is not clear why electrode layers of different materials are used (Al, Cu).
6.    Line 293. d33 should be referred to as piezoelectric constant or piezoelectric coefficient.
7.    Lines 137, 139, 155, 206, 295. References not found.

Author Response

Dear editor of Nanomaterials and reviewers:

The authors very appreciate the valuable comments from the editor and the reviewers for our manuscript entitled “Understanding the role of Soft X-Ray in charging solid-film and cellular electrets” (nanomaterial 2021270). Those comments are very helpful for revising and improving our paper, as well as the important guiding significance to our research. We have read the comments carefully and revised the manuscript thoroughly accordingly. The revision was marked up using the “Track Changes” in the manuscript, and responses to the reviewer’s comments are addressed in the attachment.

Thank you.

Yue Feng Corresponding author

School of Mechatronical Engineering Beijing Institute of Technology

Email: fengyue@bit.edu.cn

Reviewer 2 Report

The paper is devoted to development of the model of the charging effect of solid film and cellular electrets by soft X-ray radiation. This method is interesting from practical point of view in case of poling, for example, cellular electrets. The paper could be published but after some corrections:

1. Common notes: The figure should be placed after its first mention in the text.

2.In Figure 1b it will be good to show negative ions too.

3. Eq(1): symbol i means tube current. Use, please, other symbol, because later in the paper you use i as index. It could leads to misunderstanding.

4. Eqs (2), (3), (12): instead of e use, please, exp. What is e means in eq(4)?

5. In Eq(5) what is \ips_0 means? In Eq(6) what is t means?

6. In item 2.2: where is Fig.2a? I found only Fig.2b.

7. In E_pi, E_ai, etc is i means changeable index? If so separate it from indexes a, p, etc. For example in Eq(7) this leads to misunderstanding.

8. In Eq.(11) symbol A is the same as in Eq.(3) but the meanings is different. Change, please, the symbol.

9. In the text there is a lot of “Error! Reference source not found”. Starting from Eq(3). Correct, please.

10. I do not found reference to Fig.3c in the text. I think it should be before Fig 3a and 3b.      

11. In item 4.1 change, please, symbol for current as previously menthioned.

12. Before Fig.4 what k and E^n mean?

Author Response

(The authors gave the same response as above.)
